# Use of Large-Scale Genomics to Identify the Role of Animals and Foods as Potential Sources of Extraintestinal Pathogenic *Escherichia* *coli* That Cause Human Illness

**DOI:** 10.3390/foods11131975

**Published:** 2022-07-03

**Authors:** Lucas Harrison, Gregory H. Tyson, Errol Strain, Rebecca L. Lindsey, Nancy Strockbine, Olgica Ceric, Gamola Z. Fortenberry, Beth Harris, Sheryl Shaw, Glenn Tillman, Shaohua Zhao, Uday Dessai

**Affiliations:** 1U.S. Food and Drug Administration, Center for Veterinary Medicine, Laurel, MD 20708, USA; gregory.tyson@fda.hhs.gov (G.H.T.); errol.strain@fda.hhs.gov (E.S.); olgica.ceric@fda.hhs.gov (O.C.); shaohua.zhao@fda.hhs.gov (S.Z.); 2Centers for Disease Control and Prevention, Atlanta, GA 30333, USA; wmi1@cdc.gov (R.L.L.); nas6@cdc.gov (N.S.); 3U.S. Department of Agriculture, Food Safety and Inspection Service, Washington, DC 20250, USA; gamola.fortenberry@usda.gov (G.Z.F.); sheryl.shaw@usda.gov (S.S.); 4U.S. Department of Agriculture, Animal and Plant Health Inspection Service, Ames, IA 50010, USA; beth.n.harris@usda.gov; 5U.S. Department of Agriculture, Food Safety and Inspection Service, Athens, GA 30605, USA; glenn.tillman@usda.gov

**Keywords:** ExPEC, *Escherichia coli*, virulence factors, foodborne pathogens, companion animals, One Health

## Abstract

Extraintestinal pathogenic *Escherichia coli* (ExPEC) cause urinary tract and potentially life-threatening invasive infections. Unfortunately, the origins of ExPEC are not always clear. We used genomic data of *E. coli* isolates from five U.S. government organizations to evaluate potential sources of ExPEC infections. Virulence gene analysis of 38,032 isolates from human, food animal, retail meat, and companion animals classified the subset of 8142 non-diarrheagenic isolates into 40 virulence groups. Groups were identified as low, medium, and high relative risk of containing ExPEC strains, based on the proportion of isolates recovered from humans. Medium and high relative risk groups showed a greater representation of sequence types associated with human disease, including ST-131. Over 90% of food source isolates belonged to low relative risk groups, while >60% of companion animal isolates belonged to medium or high relative risk groups. Additionally, 18 of the 26 most prevalent antimicrobial resistance determinants were more common in high relative risk groups. The associations between antimicrobial resistance and virulence potentially limit treatment options for human ExPEC infections. This study demonstrates the power of large-scale genomics to assess potential sources of ExPEC strains and highlights the importance of a One Health approach to identify and manage these human pathogens.

## 1. Introduction

*Escherichia coli* is a diverse bacterial species able to adapt to a wide range of environments. *Escherichia* species are part of the normal intestinal microbiota of humans and other warm-blooded animals and can survive in many environmental reservoirs [1]. Most *E. coli* are commensal, living harmlessly within the intestinal tract of their host species, while some are pathogens capable of causing disease within or outside the intestinal tract [2]. Pathogenic strains causing disease inside the intestinal tract are referred to as intestinal pathogenic *E. coli* (IPEC), whereas strains with a propensity for causing disease outside the intestinal tract in otherwise healthy hosts are classified as extraintestinal pathogenic *E. coli* (ExPEC). ExPEC can be further classified into specialized groups, including uropathogenic *E. coli* (UPEC), which cause urinary tract infections (UTI), and neonatal meningitis *E. coli* (NMEC), which infect newborns and can cause septicemia or meningitis [3,4]. These strains can also exhibit hybrid pathotypes, encoding for virulence genes of both IPEC and ExPEC pathotypes [5].

Multilocus sequence typing (MLST) has been used to attempt to identify specific sequence types (STs) associated with ExPEC infections [6]. In a review of 217 published ExPEC studies, Manges et. al. noted that the top five clinically relevant sequence types globally (in frequency order) include ST131, ST69, ST10, ST405, and ST38. Of these, ST131 was found in over 90% of studies, and ST69 or ST10 were detected in 50% of studies [7]. However, as ExPEC are genetically heterogeneous, it is difficult to classify *E. coli* as commensals or ExPECs with ease by using traditional MLST typing alone [8]. In addition, individual *E. coli* sequence types can contain multiple pathotypes [9], further complicating associations between MLST and the ExPEC pathotype.

Several efforts have been made to better define virulence genes associated with ExPEC [6]. Recently, the Center for Genomic Epidemiology added 44 ExPEC-associated virulence genes to the *E. coli* VirulenceFinder database for the identification of ExPECs [10]. A database of *E. coli* virulence genes has also been added to AMRFinder, and these genes are automatically identified on public genome sequences [11]. However, there is no single definition for the number and type of virulence genes that designate *E. coli* as ExPEC.

ExPEC infections are important because they are responsible for millions of UTIs in the United States each year [12]. UTIs caused by ExPEC typically result from gut bacteria ascending the urethra [13]. Recent studies have implicated retail meats as a potential source of *E. coli* causing UTIs, and genomic similarities have been identified between UPEC and strains isolated from the chicken gut [14,15,16]. Prevalence-based work from retail meat sampling has also found that some ExPEC virulence genes are common in *E. coli* from retail meats [17]. Similar associations have been observed in studies evaluating *E. coli* sequence type where human UTI-associated sequence types were recovered from retail meat samples [18]. Further, in a case control study, women with antimicrobial resistant UTIs were reported to be more likely to consume chicken than those with susceptible UTIs [19]. Although no causality was established between retail meat samples and UTIs, the authors demonstrated that common genomic elements existed between these groups. The association between animals, retail meats, environment, and human UTIs is a One Health concern, since the resistance which develops in animals (non-humans) can negatively impact human health [20].

To monitor One Health antimicrobial resistance, the National Antimicrobial Resistance Monitoring System (NARMS) in the United States tracks the prevalence and resistance of foodborne pathogens in food animals, retail meats, and humans [21]. ExPEC strains also affect animal health, as demonstrated by urinary tract infections being common in non-food animals [22]. Collaborative work with the Veterinary Laboratory Investigation and Response Network (Vet-LIRN) and the National Animal Health Laboratory Network sampling includes pathogenic *E. coli* from companion animals, primarily dogs. The increase in antimicrobial resistance determinants among bacteria causing UTIs is a growing threat to both human and non-human animal health [13,23]. Infections caused by antimicrobial resistant strains of *E. coli* impose a greater clinical burden than susceptible strains, and concern is growing that resistant strains from food animals and retail meats are causative agents for a greater portion of extraintestinal infections than previously thought [24,25].

In this study, we use genomics to evaluate the potential of various *E. coli* isolation sources to harbor ExPEC strains. We compare virulence genes among strains isolated from humans to strains isolated from food and companion animals in order to identify strains that may cause ExPEC infections. This study also compares the distribution of antimicrobial resistance determinants to evaluate their potential associations with the ExPEC strains. We discuss similarities and differences in ExPEC from these various sources and how they illustrate the One Health nature of ExPEC and antimicrobial resistance (AMR).

## 2. Materials and Methods

### 2.1. Data Collection

The sequences of 41,555 candidate *E. coli* isolates and their associated metadata were collected from humans through the NARMS program by the U.S. Center for Disease Control (CDC) PulseNet (*n* = 35,621); from food animal cecal samples by the U.S. Department of Agriculture (USDA) Food Safety Inspection Service (FSIS) (*n* = 2733); from retail meats by the U.S. Food and Drug Administration (FDA) Center for Veterinary Medicine (CVM) (*n* = 2446); from dog illnesses by the CVM’s Vet-LIRN program (*n* = 663); and from other animal illnesses by the USDA’s Animal and Plant Health Inspection Service (APHIS) (*n* = 92). Publicly available sequences of *E. coli* human isolates from thirty-five academic, research, and government institutions that were not obtained through PulseNet (human non-PulseNet) supplemented our dataset. The addition of these non-PulseNet *E. coli* isolates from the NCBI database (*n* = 1268) to the NARMS dataset of 41,555 isolates brought the total number of sequences evaluated in this study to 42,823. Metadata collected included isolation source, isolation date, and collection organization, but did not include virulence phenotype or indication as a causative agent of disease. Our final dataset contained *E. coli* strains isolated from humans (*n* = 36,886); from non-human animal hosts, including cattle (*n* = 1305), swine (*n* = 738), dogs (*n* = 647), chickens (*n* = 389), turkeys (*n* = 346), cats (*n* = 30), horses (*n* = 29), uncharacterized (*n* = 6), and sheep (*n* = 1); and from retail meats of turkey (*n* = 912), chicken (*n* = 546), cattle (*n* = 526), swine (*n* = 438), and meat products whose isolation source was not characterized (*n* = 24). The above samples with an uncharacterized isolation source were labeled as untyped meat samples. The sequences collected in this study were obtained from a variety of sources, and the collection criteria for all sources was not reported. As such, trends observed in this dataset may not be representative of trends in the general *E. coli* population.

### 2.2. Strain Characterization

Strains were characterized by sequence type, phylogenetic group, and virulence type using the following libraries of indicator genes and loci. Sequence type identification was determined using MLST 2.16.1 (https://github.com/tseemann/mlst (accessed on 29 April 2020)). Sequence type for each isolate was assigned using the 7-gene Achtman multilocus sequence typing schema for *E. coli* that assigns clonal complex by the number of alleles common among related sequence types. [26]. Phylogenetic group was assigned using ClermonTyping v1.4.0 [27]. Virulence genes and antimicrobial resistance determinants for each strain were identified using AMRFinderPlus (v3.6.15 National Center for Biotechnology Information, Bethesda, MD, USA) and the VirulenceFinder database updated 05-2021 [11,28]. The results of virulence gene and AMR determinant screening for each strain were combined, and redundant hits were removed. Non-*E. coli* sequences were filtered out from the dataset using Kraken2, MLST v2.16.1 and the presence of the *ipaD* and *ipaH* genes [29,30].

IPEC were characterized with the following criteria: *stx* alleles defined STEC; the combination of *ltcA* and *stb* or *ltcA* and *sta1* defined ETEC; *eae* defined EPEC; and *aggR* defined EAEC [31]. The remaining strains were defined as the non-IPEC population. This non-IPEC population contained all strains not typed by the above criteria and included both extraintestinal pathogenic *E. coli* and commensal *E. coli*.

### 2.3. Virulence Group

Virulence gene profiles for each strain were analyzed as a presence-absence matrix (PAM) in R v3.6.2 (R Foundation for Statistical Computing, Vienna, Austria, 2021) [32]. A k-modes analysis was used to determine the virulence gene profiles best able to characterize different subpopulations of IPEC *E. coli* [33]. The optimal number of virulence gene profiles was determined by evaluating the sum of within-group differences of a series k-modes analyses allowing for 5–80 allele profiles using the elbow method [34]. Agglomerative clustering of virulence gene profiles determined virulence group relatedness. Virulence groups containing any three of the four genes *chuA*, *fyuA*, *yfcV*, and *vat* were associated with containing UPEC strains [35]. The prevalence of specific virulence gene patterns within *E. coli* isolates from humans was used to determine relative risk group categories for non-IPEC strains. Virulence groups containing >50% *E. coli* strains isolated from humans showed a strong association with a human source and were classified as having a high relative risk of containing strains that cause human disease. Virulence groups with fewer than 25% *E. coli* isolates from humans showed a weaker association with a human source and were classified as a low relative risk to human health. The remaining virulence groups containing 25–50% human isolates showed medium risk to human health relative to the other isolates, and these defined the medium relative risk groups. Due to the broad data collection methods used in this study, the effect of predicting the virulence group from a known isolation source was determined by the Goodman and Kruskal lambda value obtained through the R DescTools package v0.99.44 [36].

## 3. Results

### 3.1. Characterization of IPEC and non-IPEC Strains

To identify candidate virulence factors indicative of ExPEC strains, we first needed to identify and separate non-*E. coli* and IPEC sequences in our dataset. Screening by *Escherichia* phylogroup, the presence of *ipaD* or *ipaH*, and membership in sequence types associated with *Shigella* prompted the removal of 4791 strains from our dataset of 42,823 sequences. The resulting dataset of 38,032 strains was subdivided into two groups of 29,890 IPEC and 8142 non-IPEC. A comparison of the IPEC and non-IPEC datasets shows that 99.2% of IPEC were collected from humans through PulseNet, while 85.1% of non-IPEC strains were collected from either food animal, companion animal, or non-PulseNet human sources (Table 1).

We next evaluated the virulence gene composition of the IPEC and non-IPEC isolate collections by querying them against the AMRFinderPlus and VirulenceFinder databases. This evaluation revealed 56 virulence genes that were more than twice as prevalent in the non-IPEC strains than in the IPEC strains (Table 2). Having shown a difference in the rate of occurrence of specific virulence genes between IPEC and non-IPEC strains, we evaluated the combinations of virulence genes found among strains in the non-IPEC population.

### 3.2. Isolation Source Composition of Virulence Groups

Our k-modes analysis of virulence genes in the non-IPEC population defined 40 groups of non-IPEC strains with unique virulence gene profiles (Figure 1). The size of each virulence group ranged from 23 to 1026 isolates, and groups contained 2–34 virulence genes (Appendix A). Each group was defined by a pattern of virulence genes present in >70% of strains in the group. The proportion of strains isolated from humans within each virulence group was used to characterize the groups as having a low, medium, or high relative risk of containing ExPEC strains (Appendix A). Twelve virulence groups contained ≥50% human isolates and were considered to have a high relative risk of containing ExPEC strains: groups 3, 5, 7, 12, 13, 14, 16, 17, 26, 33, 35, and 36 with a combined population of 1743 isolates. None of the virulence genes were conserved among all the high relative risk virulence groups. Seven virulence genes were exclusive to the high relative risk groups: *sat*, *sfaS*, *iucD*, *iucB*, *sta1*, *capU*, and *nfaE*. Five virulence groups contained from 25% to 50% human isolates, consisted of 845 strains, and were classified as having a medium risk of containing ExPEC strains. The remaining 5554 strains belonged to 23 virulence groups. These 23 virulence groups contained ≤25% strains isolated from humans and were considered to have a low relative risk of containing ExPEC strains.

Source composition of the non-human fraction of isolates from FSIS-NARMS, CVM-NARMS, APHIS and Vet-LIRN showed that isolates were not found in equal ratios across the virulence relative risk groups (Appendix A). While strains from all non-human sources were most prevalent in the low relative risk categories, companion animal isolates were more likely to be found in the medium and high relative risk groups compared to other non-human sources. (Figure 2). Evaluation of the dataset by calculating the Goodman and Kruskal lambda returned a lambda value of 0.295 for informing the prediction of relative risk group, given the isolation source.

We then evaluated the composition of the relative risk groups by their *E. coli* phylogenetic groups (Appendix A). The major phylogenetic groups consisting of at least 20% of the relative risk groups were A, B1, B2, and D (Table 3). In the high relative risk group, 56.2% of the isolates belonged to B2, and 20.4% belonged to the D phylogenetic group. Among the medium relative risk group, 59.3% of isolates belonged to B2, and 20.2% belonged to phylogenetic group A. Phylogenetic group B1 was most common in the low relative risk group, at 48.1%, followed by phylogenetic group A, at 28.6%. In the remaining phylogenetic groups, groups C and G were most common in the low relative risk group, at 3.8% and 3.7%, respectively, while groups E and F were found most often in the high relative risk group, at 4.6% and 4.7%, respectively. A phylogenetic group comparison to virulence group revealed two main clusters of high relative risk virulence groups. In the first cluster, more than 70% of the isolates from high relative risk groups 26, 12, 33, 35, and 16 belonged to phylogenetic group B2. The second cluster of high relative risk groups of 5, 13, 14, 17, and 3 were represented by phylogenetic groups E, D, and A.

### 3.3. Sequence Type and AMR Gene Composition of Virulence Groups

Seven-gene multi-locus sequence typing analysis identified 1031 sequence types in our dataset of 8142 non-IPEC isolates. Of the 1031 sequence types, 194 sequence types belonged to 43 clonal complexes, accounting for 4139 isolates. The remaining 837 sequence types classified 3710 isolates. The final set of 293 strains did not match any sequence type in the PubMLST database. Sequence types contained multiple virulence gene profiles and were distributed among the 40 virulence groups (Figure 3). While no virulence group was exclusive to a specific sequence type, each of the virulence groups contained a sequence type that represented at least 25% of its strains (Appendix A). The majority of strains within eight virulence groups belonged to a single sequence type: within virulence group 5, 89.5% of the strains belonged to ST182; 85.7% of strains from virulence group 17 belonged to the ST38 clonal complex; 84.4% of virulence group 26, 80.3% of virulence group 12, and 54.9% of virulence group 33 belonged to the ST131 clonal complex; 82.1% of virulence group 15 belonged to ST117; 68.9% of virulence group 3 belonged to the ST10 clonal complex; and 51.2% of strains from virulence group 39 belonged to the ST23 clonal complex.

Every virulence group contained multiple sequence types, and 39/40 virulence groups were contained in more than one isolation source. We then subdivided the virulence groups by sequence type to determine if the proportion of human isolates was consistent for all sequence types within a virulence group. Of the 572 sequence type/virulence group combinations in our dataset that contained human isolates, 82 sequence type/virulence group combinations had 5 or greater strains isolated from humans and 21 sequence type/virulence group combinations contained only human isolates (Appendix A). In 4 of the 21 combinations, the sequence type containing human isolates accounted for all the human isolates in the virulence group. In the remaining 17 of the 21 sequence type-virulence group combinations, the combinations identified sequence types among 6 virulence groups that only contained human isolates.

Twelve virulence groups contained virulence gene patterns associated with UPEC isolates [35]: 8, 12, 15, 16, 26, 30, 32, 33, 35, 37, 38, and 40. The 12 UPEC-associated virulence groups accounted for 25.3% of the non-IPEC isolates, or only 5.4% of our combined IPEC and non-IPEC dataset. Isolates belonging to these UPEC-associated virulence groups represented 24.4% of our non-IPEC PulseNet isolates and 64.05% of non-IPEC human isolates obtained from sources other than PulseNet. The distribution of strains from UPEC-associated virulence groups isolated from non-human sources accounted to 16.2% isolates from retail meats, 3.9% isolates from food animals, and 64.5% isolates from companion animals. Isolates in dogs were common in 5/12 of the UPEC-associated virulence groups and ranged from a low of 26.6% in group 16, up to 74% in group 31. Fewer than 3.8% of the strains isolated from these 5 virulence groups were isolated from food animals or retail meats.

The AMR profiles of the 40 virulence groups were evaluated by their relative risk group. We found that 18 out of the 26 most common AMR determinants were found more than twice as often in the high relative risk ExPEC virulence groups compared with the other groups (Figure 4). Of these 18 AMR determinants, mutations causing substitutions *gyrA*(S83L), *gyrA*(D87N), *parC*(S80I), *parC*(E84V), *parE*(I529L), and *marR*(S3N) were found in companion animals at a >2-fold higher rate than in the other non-human sources (Table 4). Additionally, the AMR genes *mph(A)*, *dfrA17*, *aadA5*, *bla*_CTX-M-15_, *bla*_OXA-1_, and *catB3* were found in companion animals at a >2-fold higher rate than in the other non-human sources. In total, 12/18 AMR determinants more commonly found in the high relative risk virulence groups were isolated from companion animals at a higher rate than from any other non-human source.

## 4. Discussion

ExPEC are a threat to human health, causing millions of urinary tract and other extraintestinal infections in the United States each year [37]. However, the sources of ExPEC are not always known, nor are the precise combinations of genes necessary for pathogenicity. Our study used genomics data from various human and non-human animal sources to initially aid in understanding the markers for ExPEC, and then to assess their relative distributions in retail meat, food animals, companion animals, and humans.

Of note, only a few *E. coli* isolated from food animals and retail meats were in the high and medium relative risk ExPEC categories. Analyzed individually, none of the food animal or retail meat sources had a >15% representation of high and medium relative risk ExPEC strains. This was contradictory to our expectations, as previous work with chicken and human isolates had suggested a strong association between chicken and human ExPEC pathotypes, while our analysis showed less than 5% of non-IPEC isolates from chicken belonged to the high relative risk groups [14]. *E. coli* strains isolated from retail meats, however, were common in three of the UPEC-associated virulence groups. Nevertheless, only 8% of isolates from food animals and retail meats were in the medium and high-relative risk ExPEC groups. Despite the lower relative presence of the higher relative risk groups in food animals and retail meats, their potential as sources of ExPEC risk to humans cannot be completely discounted.

Our approach of grouping ExPEC strains by virulence gene profiles is helpful to further differentiate existing classification schema, such as sequence typing or phylogenetic grouping (Appendix A). For example, the high relative risk strains of the broad host range ST10 complex (Cplx) were differentiated from the lower relative risk ST10 Cplx strains by their virulence gene composition. Grouping the dataset by virulence genes showed utility for identifying strains that may pose a threat to animal health, which can be seen in virulence group 15, being made up of ST117 *E. coli*. The ST117 sequence type is known to contain avian pathogenic *E. coli*.

In addition to aiding in the analysis of sequence types, the virulence group subdivision of phylogenetic groups highlighted which strains may be of greater concern than did the classification by phylogenetic grouping alone. While the phylogenetic group B2 is often associated with ExPEC infections, our division by virulence groups classified B2 subsets into low, medium, and high relative risk categories. Our virulence groups also identified a subset of phylogenetic group A strains with a high relative risk of containing ExPEC strains as virulence group 3.

A noteworthy finding was that the proportion of 18 out of 26 resistance determinants was higher among ExPEC in the high relative risk group than in the low relative risk group (Figure 3). This unequal distribution of AMR genes among relative risk groups can be concerning, since ExPEC infections typically require antimicrobial treatment [38]. Given the similarity of human isolates to companion animal isolates, this may warrant further investigation into the directionality of AMR and pathogen transmission between owners and companion animals. Further, since antimicrobial use in food animals can provide selective pressure for AMR, the presence of high relative risk ExPEC strains in food animals can negatively impact human health [39,40]. It is important to note that, in this genomic study, we did not perform any assessment of antimicrobial use and its impact on AMR phenotypes.

Although this study is the largest reported work to date using genomics to assess virulence and AMR in ExPEC, it did have some limitations. For instance, we did not perform virulence assays, so our virulence groups are not validated by phenotypic measures. Thus, the virulence genes found in the high relative risk ExPEC group may not be the most important genes contributing to ExPEC phenotypes. However, even if the genes are only correlated with ExPEC infections, their high rate of occurrence among strains causing clinical illness allows them to be used as indicators for the relative risk of human infection. Moreover, the removal of IPEC isolates was necessary to focus on ExPEC, but we may have eliminated some isolates with IPEC/ExPEC hybrid phenotypes [5]. Another limitation is that the human ExPEC isolates were not collected in a nationally representative and systematic surveillance system, meaning there could be bias in the types and numbers of ExPEC represented in the study. This may have contributed to a greater representation of AMR determinants in isolates of higher-virulence groups, since bacteria with treatment failures may have been more likely to be collected and sequenced. This sampling contrasts with the other isolation sources, which were collected as part of routine surveillance in the United States. An additional limitation of this study is the difference in population size and source composition among the virulence groups and isolation sources. This need for increased data diversity illustrates the importance of collaborative data collection efforts among organizations that represent unique interactions between animal and human health.

This study highlights the power of large-scale genomics and diverse data sources in addressing important One Health questions, particularly those concerning the relationship between antimicrobial resistance and virulence in anthropozoonotic pathogens. We used our large collection of sequencing data from different sources, generated by five federal organizations, to gain an understanding of genes that are linked to extraintestinal infections. Our results show that most *E. coli* from food animals and retail meats are not in the high-risk ExPEC groups. However, a large portion of non-IPEC strains have an increased potential of containing ExPEC strains, as these share virulence genes with isolates causing human or animal illnesses. Further, the contribution of AMR in ExPEC strains can lead to difficult to treat and more serious infections. This is a critical area for future research.

We believe that this unique approach using large-scale genomics on a diverse ExPEC source dataset to arrive at potential isolation sources without the biases of sequence type or multidrug resistance markers was essential to understanding ExPEC in the context of the burden of human illness. With additional datasets and analyses, this approach can further our understanding of ExPEC strains and UPEC pathotypes. This approach can also be applied more broadly to complex and difficult to decipher human-animal disease systems to gain an in-depth understanding of the agents, their roles in disease development, and their risk potential, as well as to predict impacts on human and animal health.

## Figures and Tables

**Figure 1 foods-11-01975-f001:**
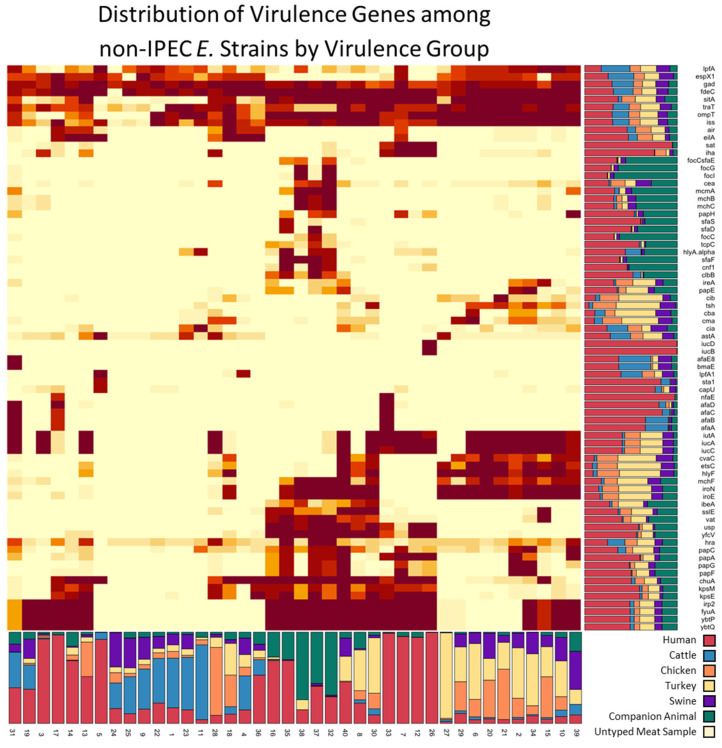
Heatmap of virulence genes organized by virulence groups from the 8142 non-IPEC strains. Stacked bar plots below and to the right of the heatmap show the relative contribution of each isolation source to the virulence groups and virulence genes, respectively. Animal isolation sources are the combined results from all contributing organizations.

**Figure 2 foods-11-01975-f002:**
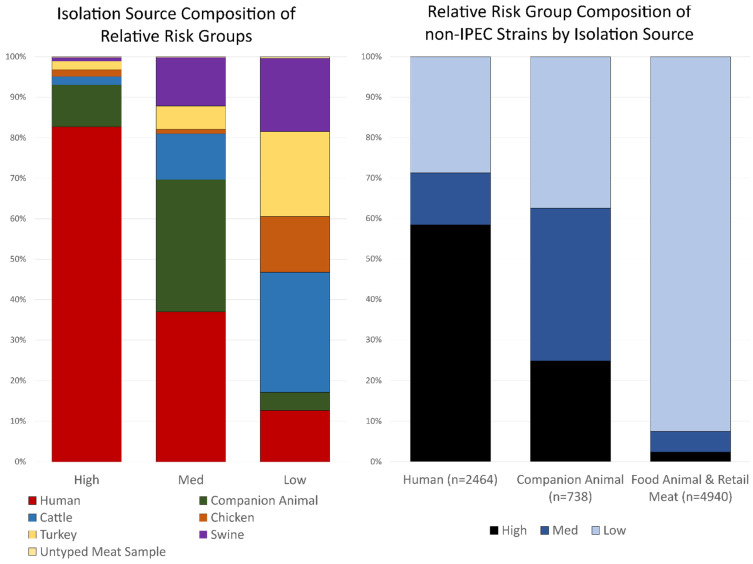
(**left**) Bar graph representation of isolation sources divided into relative risk groups. Isolation sources are the combined results from all contributing organizations. (**right**) Evaluation of the relative risk group composition of strains from humans (PulseNet and non-PulseNet), companion animals (Vet-LIRN and APHIS), and food animals + retail meats (USDA-FSIS and CVM-NARMS).

**Figure 3 foods-11-01975-f003:**
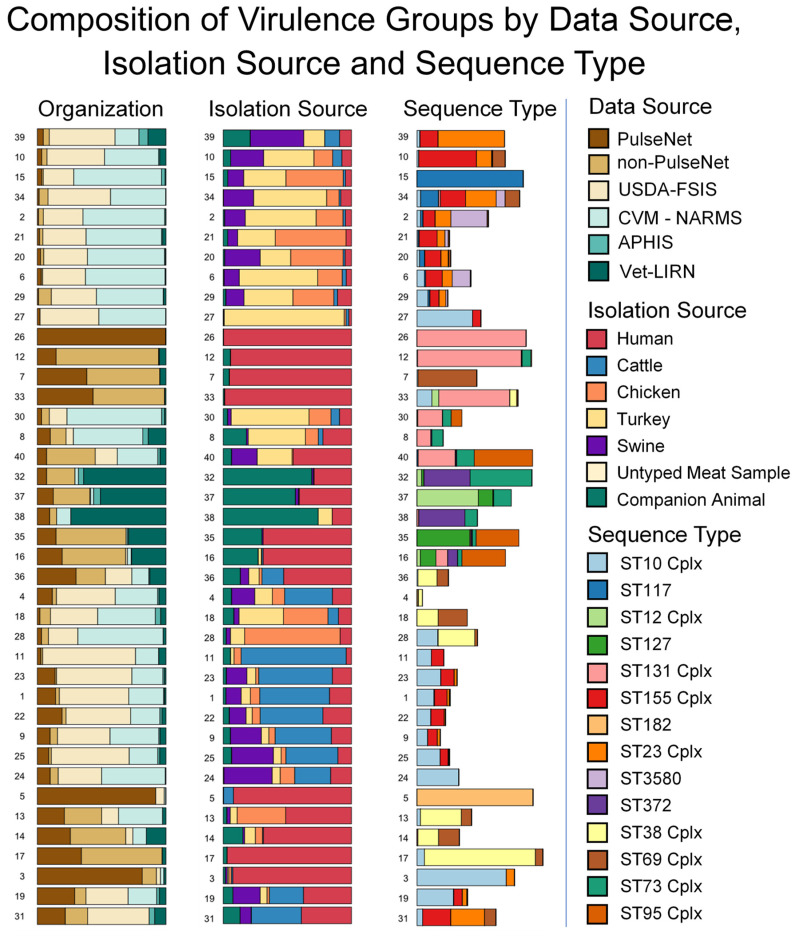
Isolation source composition of, and sequence type distribution among, non-IPEC virulence groups. Cplx designation of sequence type indicates a clonal complex. Virulence group order is consistent with that in Figure 1, to aid visual comparison. Source data can be found in Appendix A. Animal isolation sources are the combined results from all contributing organizations.

**Figure 4 foods-11-01975-f004:**
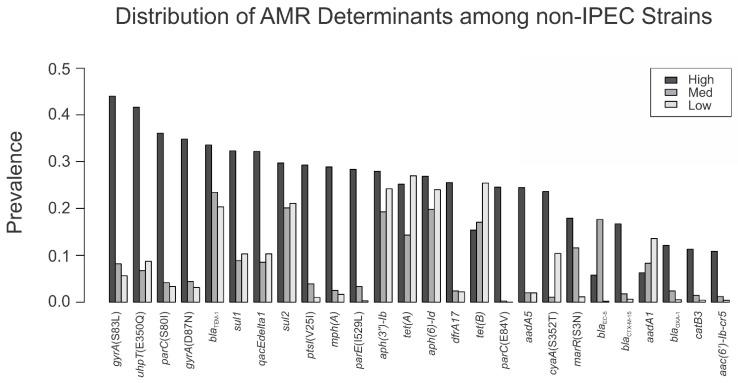
The prevalence of AMR determinants in at least 10% of any of the relative risk groups shows that 18/26 of this set of AMR determinants are present in the high relative risk group at a 2-fold or greater frequency than in either the medium or low relative risk groups.

**Table 1 foods-11-01975-t001:** Distribution of non-IPEC strains by isolation source.

	# of Isolates	IPEC	Non-IPEC
**Human (PulseNet)**	30,862	29,651	1211
**Human (non-PulseNet)**	1268	15	1253
**Retail meats**	2433	31	2402
**Food animal cecal**	2717	179	2538
**Companion animals**	752	14	738
**Total**	38,032	29,890	8142

The # symbol indicates we are describing the quantity, or number of isolates.

**Table 2 foods-11-01975-t002:** Virulence genes with >2-fold greater representation in the non-IPEC dataset.

Virulence Gene	IPEC	Non-IPEC	Non-IPEC/IPEC
*tsh*	2.78%	14.01%	5.046
*etsC*	5.12%	25.35%	4.954
*clbB*	1.89%	9.36%	4.945
*vat*	3.55%	17.56%	4.941
*cnf1*	1.55%	7.61%	4.911
*hlyF*	5.61%	27.14%	4.842
*f17A*	0.75%	3.61%	4.817
*iroE*	6.98%	33.39%	4.785
*sfaF*	1.59%	7.58%	4.759
*iroN*	7.06%	33.54%	4.751
*focI*	0.97%	4.58%	4.742
*papH*	1.59%	7.53%	4.721
*tcpC*	1.26%	5.96%	4.721
*f17G*	0.72%	3.39%	4.697
*focG*	0.88%	4.09%	4.636
*papA*	4.77%	21.94%	4.594
*usp*	4.99%	22.92%	4.592
*focC*	0.77%	3.45%	4.506
*papF*	3.45%	15.45%	4.478
*kpsM*	6.37%	28.21%	4.429
*papC*	5.88%	25.88%	4.402
*ibeA*	1.95%	8.47%	4.339
*sfaD*	0.88%	3.75%	4.246
*papE*	2.09%	8.86%	4.231
*sslE*	3.64%	15.35%	4.217
*cvaC*	4.87%	20.40%	4.189
*iucD*	0.12%	0.49%	4.138
*iucB*	0.12%	0.50%	4.081
*yfcV*	6.25%	24.80%	3.967
*nfaE*	0.50%	1.98%	3.951
*sfaS*	0.54%	2.08%	3.843
*sfaE*	0.21%	0.81%	3.785
*lngA*	0.25%	0.92%	3.733
*kpsE*	8.46%	31.45%	3.720
*air*	2.44%	8.90%	3.653
*afaE*	0.29%	1.04%	3.646
*ltcA*	0.41%	1.49%	3.586
*hlyE*	0.22%	0.77%	3.574
*eilA*	3.36%	11.61%	3.450
*eatA*	0.50%	1.72%	3.435
*mchF*	8.09%	27.61%	3.413
*sat*	3.05%	9.90%	3.241
*hra*	9.97%	30.19%	3.029
*iroD*	0.02%	0.06%	2.931
*afaB*	1.15%	3.33%	2.906
*afaA*	1.17%	3.34%	2.853
*cma*	5.63%	14.75%	2.618
*afaC*	1.01%	2.58%	2.541
*ccI*	0.13%	0.32%	2.407
*mchB*	3.59%	8.65%	2.406
*afaD*	1.64%	3.93%	2.398
*faeG*	0.02%	0.05%	2.345
*ireA*	3.99%	9.31%	2.333
*iroB*	0.02%	0.04%	2.261
*iroC*	0.02%	0.04%	2.261
*neuC*	4.04%	8.29%	2.054

**Table 3 foods-11-01975-t003:** Distribution of relative risk group by *E. coli* phylogenetic grouping.

	A	B1	B2	C	D	E	E or Clade I	F	G	Unknown
**High**	10.8	0.9	56.2	0.6	20.4	4.6	0.2	4.7	1.3	0.2
**Med**	20.2	17	59.3	2.4	0.5	0.4	0	0.1	0.1	0
**Low**	28.6	48.1	7.6	3.8	4	2.7	0	1.2	3.7	0.3

Values represented as percent of isolates from each relative risk group belonging to the phylogenetic group.

**Table 4 foods-11-01975-t004:** Distribution of AMR determinants associated with high relative risk strains among isolation sources.

Resistance Determinant	Human	Cattle	Chicken	Turkey	Swine	Untyped Meat Sample	Companion Animal
*cyaA*(S352T)	15.1	2.5	32.6	14.9	5.4	43.5	8.9
*gyrA*(S83L)	37.2	2.3	2.2	1.7	3.5	4.3	15.5
*parC*(S80I)	28.9	0.5	0.3	0.8	2.3	0.0	12.4
*gyrA*(D87N)	27.7	0.4	0.6	0.8	2.1	0.0	12.2
*uhpT*(E350Q)	29.6	6.9	12.7	13.3	4.1	8.7	13.8
*mph(A)*	22.1	0.2	0.1	0.2	2.0	0.0	5.8
*dfrA17*	19.6	0.6	0.3	1.2	2.1	0.0	7.5
*parE*(I529L)	19.3	0.0	0.1	0.8	1.9	0.0	3.9
*ptsI*(V25I)	20.1	0.1	1.8	2.7	2.0	4.3	4.0
*aadA5*	18.5	0.5	0.1	1.3	1.4	0.0	7.4
*parC*(E84V)	16.8	0.0	0.0	0.0	0.0	0.0	2.4
*marR*(S3N)	13.3	0.7	1.4	1.0	0.2	4.3	14.1
*bla* _CTX-M-15_	12.8	0.1	0.0	0.1	0.1	0.0	3.1
*bla* _OXA-1_	9.6	0.0	0.0	0.4	0.0	0.0	2.6
*catB3*	8.7	0.1	0.0	0.0	0.0	0.0	2.4
*aac(6′)-Ib-cr5*	8.2	0.0	0.0	0.0	0.0	0.0	2.7
*sul1*	28.0	2.8	20.5	14.8	5.8	21.7	7.3
*qacEdelta1*	27.9	2.7	20.4	14.6	5.8	21.7	7.8

Values represented as percent of isolates from source with AMR determinants.

## Data Availability

The data presented in this study are available in this article and in the Appendix A.

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
