# Peer review of "Use of Large-Scale Genomics to Identify the Role of Animals and Foods as Potential Sources of Extraintestinal Pathogenic Escherichia coli That Cause Human Illness"

_foods, 2022, doi:10.3390/foods11131975_

Round 1
Reviewer 1 Report
Comments to the authors for evaluating the following manuscript
Title:
Use of Large-Scale Genomics to Identify the Role of Animals 2 and Foods as Potential Sources of Extraintestinal Pathogenic Escherichia coli that Cause Human Illness
Title: the title is too long and it should be rephrased
Keywords: Key words of the manuscript are too long and uninformative, you must concise them.
Abstract section
· Please add the overall practical implementation of your results or other hypothesis that may be utilized in the future at the end of the abstract. Such as the over aim for the rapid detection.
Introduction section
· The sentences in the introduction section are too long and need rephrase, it isnot recommended to add one sentence in 7 lines “37:43”
· Throughout the manuscript, many punctuations were incorrect, please carefully revise, for example “line 51”
· Please if you can to elaborate in detail the necessary for the rapidly detection of ExPEC in the clinical fields
· Can you add a paragraph about the role of animal in the transmission of ExPEC
Methodology:
· Can you provide the tested antimicrobial agents and virulence genes, primers, conditions in the main text or supplementary date
Results
· The results should be divided into several sections as in the methodology
· Please provide the accession number for the sequenced genes in MLST
· Table 2 is too long, therefore you have to replace it with simple figures
· The scientific written of the abbreviation of the tested genes throughout the manuscript need to be revised “When referring to the genetic element (or genotype), the abbreviation must be italicized and the first 3 letters are lowercase”
Discussion:
· It needs numerous modifications. It should focus on explaining and evaluating what you found (the main results), showing how it relates to the new researches
Conclusion section:
· It must be rephrased, conclusion section must provide us with the applied implication of your results in concise manner
· Please add at the end of the manuscript the limitations: what can’t the results and discussion tell us?
Author Response
Title
Comment: Title: the title is too long and it should be rephrased
Response: We recognize that the manuscript has a long title. Unfortunately, this title has undergone multiple rounds of institutional review and altering it at this stage will require initiation of a new approval process. We would request to keep the original title.
Comment: Keywords: Key words of the manuscript are too long and uninformative, you must concise them.
Response: We agree that uninformative keywords are to be avoided. Because we did not evaluate phenotypic virulence in these strains, we have removed the ‘virulence’ keyword
Abstract section
Comment: Please add the overall practical implementation of your results or other hypothesis that may be utilized in the future at the end of the abstract. Such as the over aim for the rapid detection.
Response: We agree that this is important information, and the aim of the manuscript is to demonstrate how analyses of large datasets can be used to inform potential sources of exposure of humans to ExPEC infections. We believe that the final sentence of the abstract makes this point.
Regarding the reviewer’s comment regarding rapid detection, we are not proposing a method for the rapid detection of ExPEC. While this is certainly a clinically-relevant topic, it is outside the scope of this manuscript.
Introduction section
Comment: The sentences in the introduction section are too long and need rephrase, it isnot recommended to add one sentence in 7 lines “37:43”
Response: Lines 37:43 are two sentences, however the formatting at the end of line 40 may have obscured the end of the sentence.
Comment: Throughout the manuscript, many punctuations were incorrect, please carefully revise, for example “line 51”
Response: We have reviewed the manuscript and carefully modified the punctuations.
Comment: Please if you can to elaborate in detail the necessary for the rapidly detection of ExPEC in the clinical fields
Response: We agree that the rapid detection of ExPEC strains in a clinical setting is an important topic, and worthy of further investigation. However, this is not a topic that this manuscript was designed to evaluate. Our goal was to evaluate existing sequence data and identify which isolation sources likely contribute to human infection. In an effort to ensure the reader does not think we are proposing a detection method, we changed the term in line 57 to ‘identification’ instead of ‘detection’.
Comment: Can you add a paragraph about the role of animal in the transmission of ExPEC
Response: We agree that this information is very important and relevant to the manuscript. The fourth paragraph of the Introduction addresses the role of retail meats in the transmission of ExPEC. The fifth paragraph of the Introduction addresses the association between food animals and companion animals regarding the transmission of ExPEC.
Methodology:
Comment: Can you provide the tested antimicrobial agents and virulence genes, primers, conditions in the main text or supplementary date
Response: We agree that it is very important to communicate these parameters when discussing work done with live organisms. However, our work was solely bioinformatic, and no live samples were evaluated in the study. As such, no antimicrobial agents, growth conditions or primers were used. Regarding virulence genes, our Materials and Methods sections cites the VirulenceFinder and AMRFinder databases as our reference dataset of virulence genes.
Results
Comment: The results should be divided into several sections as in the methodology
Response: We agree that this would facilitate read accesses, and headers have been added to the appropriate sections of the Results.
Comment: Please provide the accession number for the sequenced genes in MLST
Response: We agree that this is important information and we have added the accession numbers for the isolates analyzed by MLST as supplementary data.
Comment: Table 2 is too long, therefore you have to replace it with simple figures
Response: We agree that replacing a table with a figure is valid option when a table contains multiple comparisons. Table 2 details the ratio of specific virulence genes between two populations. Unfortunately, converting the table into a figure would not reduce the number of elements displayed, and we believe that the table in its present format is appropriate.
Comment: The scientific written of the abbreviation of the tested genes throughout the manuscript need to be revised “When referring to the genetic element (or genotype), the abbreviation must be italicized and the first 3 letters are lowercase”
Response: Proper scientific notation is important. In the original draft, we did not italicize the genes associated with mutations that result amino acid substitutions. We agree with the reviewer’s comments and have changed the formatting of these genes and their associated amino acid substitutions.
Discussion:
Comment: It needs numerous modifications. It should focus on explaining and evaluating what you found (the main results), showing how it relates to the new researches
Response: We propose that our Discussion summarizes our findings, highlights the interesting results and presents them in the context of current research as described below:
- Discussion Paragraph 1: Summary of the project
- Discussion Paragraph 2: Evaluates potential for virulence against isolation source and compares to current research in the field
- Discussion Paragraph 3: Compares results of virulence gene grouping to standard sequence typing methods currently used
- Discussion Paragraph 4: Compares results of virulence gene grouping to standard Escherichia phylogenetic grouping
- Discussion Paragraph 5: Highlights an interesting association between AMR genes and virulence
- Discussion Paragraph 6: Details some of the limitations of the study
- Discussion Paragraph 7: Conclusion paragraph
We believe that the paragraphs as written address the reviewer’s comments. However, if the reviewer has additional comments, we would be willing to address them.
Conclusion section:
Comment: It must be rephrased, conclusion section must provide us with the applied implication of your results in concise manner
Response: We agree with the reviewer that the conclusions need to be presented in a concise format. We recognized that the concluding paragraph contain two main concepts and have divided the paragraph into two. The second of these paragraphs contains the implication of our results to the field.
Comment: Please add at the end of the manuscript the limitations: what can’t the results and discussion tell us?
Response: We agree that recognizing the limitations of a study is an important part of the writing process. The requested limitations have been addressed in the sixth paragraph of the Discussion, lines 357-376.
Reviewer 2 Report
The manuscript entitled “Use of Large-Scale Genomics to Identify the Role of Animals and Foods as Potential Sources of Extraintestinal Pathogenic Escherichia coli that Cause Human Illness” is well designed, written and structured by Harrison et al., strongly appropriate English with real clear structure. They implemented a valuable big genomics data analysis to evaluate the potential sources of extra-intestinal pathogenic E. coli infections by using the genomic data of E. coli isolates from five U.S. government organizations. The results are so interesting, they implemented the genomics analysis for 41555 E. coli isolated from food, human and animal samples. however, there are some major concerns in this study which should be addressed; consequently, the quality of the paper will be improved according to the scientific standards of the Foods journal.
- You must include your list of the strains (all 41555 isolates and the source of each isolate) with accession numbers, if accessible, in an excel file as supplementary data to the manuscript.
- Please include the statistical analysis section in the materials and methods completely, the software and statistical packages you used.
- Plasmids are important in ExPEC strains and other foodborne pathogens and can be detected in bacterial genomes by using PlasmidFinder in-silico tools. Please add this analysis to your manuscript and update all sections accordingly.
Author Response
Comment: You must include your list of the strains (all 41555 isolates and the source of each isolate) with accession numbers, if accessible, in an excel file as supplementary data to the manuscript.
Response: We recognize that this is important information and the requested file has been added to the Supplementary Materials
Comment: Please include the statistical analysis section in the materials and methods completely, the software and statistical packages you used.
Response: This request highlights one of the limitations of the study. The data collection methods were not standardized across data repositories, as cited in the Discussion. Because of this, the only appropriate statistical test we could perform was evaluating the Goodman-Kruskal lambda value. This test was performed in R with the DescTools v0.99.44 package, and this information has been added to the manuscript.
Comment: Plasmids are important in ExPEC strains and other foodborne pathogens and can be detected in bacterial genomes by using PlasmidFinder in-silico tools. Please add this analysis to your manuscript and update all sections accordingly.
Response: We recognize the important role that plasmids have in bacterial pathogenesis and antimicrobial resistance. We did not include a plasmid analysis due to a limitation inherent in the available data. Briefly, our dataset contained draft assemblies that organize the sequence in many contigs. Current in silico plasmid tools such as PlasmidFinder only identify the contig that contains the plasmid replicon. In light of our analysis of virulence genes in the manuscript, the limitation described above would prevent us from attributing a given virulence gene to a specific plasmid type. We agree that including a distribution of plasmid type among the virulence group populations would be interesting. Unfortunately, this analysis would also invite the reader to infer causal relationships between plasmid type and virulence genes that the data do not support.
Round 2
Reviewer 1 Report
thanks for the well correction and mangement all my criticisms and i have no other comments for this manuscript
Reviewer 2 Report
All revisions have been addressed and the manuscript can now be accepted for publication in its present form.